# MODELLING THE INFLUENCE OF DATA STRUCTURE ON LEARNING IN NEURAL NETWORKS

## ABSTRACT

The lack of crisp mathematical models that capture the structure of real-world data sets is a major obstacle to the detailed theoretical understanding of deep neural networks. Here, we first demonstrate the effect of structured data sets by experimentally comparing the dynamics and the performance of two-layer networks trained on two different data sets: (i) an unstructured synthetic data set containing random i.i.d. inputs, and (ii) a simple canonical data set containing MNIST images. Our analysis reveals two phenomena related to the dynamics of the networks and their ability to generalise that only appear when training on structured data sets. Second, we introduce a generative model for data sets, where high-dimensional inputs lie on a lower-dimensional manifold and have labels that depend only on their position within this manifold. We call it the *hidden manifold model* and we experimentally demonstrate that training networks on data sets drawn from this model reproduces both the phenomena seen during training on MNIST.

## 1   INTRODUCTION AND RELATED WORK

A major impediment for understanding the effectiveness of deep neural networks is our lack of mathematical models for the data sets on which neural networks are trained. This lack of tractable models prevents us from analysing the impact of data sets on the training of neural networks and their ability to generalise from examples, which remains an open problem both in statistical learning theory (Vapnik, 2013; Mohri et al., 2012), and in analysing the average-case behaviour of algorithms in synthetic data models (Seung et al., 1992; Engel & Van den Broeck, 2001; Zdeborová & Krzakala, 2016).

Indeed, most theoretical results on neural networks do not model the structure of the training data, while some works build on a setup where inputs are drawn component-wise i.i.d. from some probability distribution, and labels are either random or given by some random, but fixed function of the inputs. Despite providing valuable insights, these approaches are by construction blind to key structural properties of real-world data sets.

Here, we focus on two types of data structure that can both already be illustrated by considering the simple canonical problem of classifying the handwritten digits in the MNIST database using a neural network $\mathcal{N}$ (LeCun & Cortes, 1998). The input patterns are images with $28 \times 28$ pixels, so *a priori* we work in the high-dimensional $\mathbb{R}^{784}$. However, the inputs that may be interpreted as handwritten digits, and hence constitute the "world" of our problem, span but a lower-dimensional manifold within $\mathbb{R}^{784}$ which is not easily defined. Its dimension can nevertheless be estimated to be around $D \approx 14$ based on the neighbourhoods of inputs in the data set (Grassberger & Procaccia, 1983; Costa & Hero, 2004; Levina & Bickel, 2004; Facco et al., 2017; Spigler et al., 2019). The intrinsic dimension being lower than the dimension of the input space is a property expected to be common to many real data sets used in machine leanring. We should not consider presenting $\mathcal{N}$ with an input that is outside of its world (or maybe we should train it to answer that the "input is outside of my world" in such cases). We will call inputs *structured* if they are concentrated on a lower-dimensional manifold and thus have a lower-dimensional latent representation.

The second type of the structure concerns the function of the inputs that is to be learnt, which we will call the learning task. We will consider two models: the *teacher task*, where the label is obtained as a function of the high-dimensional input; and the *latent task*, where the label is a function of only the lower-dimensional latent representation of the input.

| | |
|---|---|
| structured inputs | inputs that are concentrated on a fixed, lower-dimensional manifold in input space |
| latent representation | for a structured input, its coordinates in the lower-dimensional manifold |
| task | the function of the inputs to be learnt |
| latent task | for structured inputs, labels are given as a function of the latent representation only |
| teacher task | for all inputs, labels are obtained from a random, but fixed function of the high-dimensional input without explicit dependence on the latent representation, if it exists |
| MNIST task | discriminating odd from even digits in the MNIST database |
| vanilla teacher-student setup | Generative model due to Gardner & Derrida (1989), where data sets consist of component-wise i.i.d. inputs with labels given by a fixed, but random neural network acting directly on the input |
| hidden manifold model (HMF) | Generative model introduced in Sec. 4 for data sets consisting of structured inputs (Eq. 6) with latent labels (Eq. 7) |

Table 1: Several key concepts used/introduced in this paper.

We begin this paper by comparing neural networks trained on two different problems: the *MNIST task*, where one aims to discriminate odd from even digits in the in the MNIST data set; and the *vanilla teacher-student setup*, where inputs are drawn as vectors with i.i.d. component from the Gaussian distribution and labels are given by a random, but fixed, neural network acting on the high-dimensional inputs. This model is an example of a teacher task on unstructured inputs. It was introduced by Gardner & Derrida (1989) and has played a major role in theoretical studies of the generalisation ability of neural networks from an average-case perspective, particularly within the framework of statistical mechanics (Seung et al., 1992; Watkin et al., 1993; Engel & Van den Broeck, 2001; Zdeborová & Krzakala, 2016; Advani & Saxe, 2017; Aubin et al., 2018; Barbier et al., 2019; Goldt et al., 2019; Yoshida et al., 2019), and also in recent statistical learning theory works, e.g. (Ge et al., 2017; Li & Y., 2017; Mei & Montanari, 2019; Arora et al., 2019). We choose the MNIST data set because it is the simplest widely used example of a structured data set on which neural networks show significantly different behaviour than when trained on synthetic data of the vanilla teacher-student setup.

Our reasoning then proceeds in two main steps:

1. *We experimentally identify two key differences between networks trained in the vanilla teacher-student setup and networks trained on the MNIST task (Sec. 3).* i) Two identical networks trained on the same MNIST task, but starting from different initial conditions, will achieve the same test error on MNIST images, but they learn globally different functions. Their outputs coincide in those regions of input space where MNIST images tend to lie – the "world" of the problem, but differ significantly when tested on Gaussian inputs. In contrast, two networks trained on the teacher task learn the same functions globally to within a small error. ii) In the vanilla teacher-student setup, the test error of a network is stationary during long periods of training before a sudden drop-off. These plateaus are well-known features of this setup (Saad & Solla, 1995; Engel & Van den Broeck, 2001), but are not observed when training on the MNIST task nor on other datasets used commonly in machine learning.

2. **Our main contribution:** *We introduce the* hidden manifold model *(HMF), a probabilistic model that generates data sets containing high-dimensional inputs which lie on a lower-dimensional manifold and whose labels depend only on their position within that manifold (Sec. 4).* In this model, inputs are thus structured and labels depend on their lower-dimensional latent representation. We experimentally demonstrate that training networks on data sets drawn from this model reproduces both behaviours observed when training on MNIST. We also show that the structure of both, input space and the task to be learnt, play an important role for the dynamics and the performance of neural networks.

**Other related work** Several works have compared neural networks trained from different initial conditions on the same task by comparing the different features learnt in vision problems (Li et al., 2015; Raghu et al., 2017; Morcos et al., 2018), but these works did not compare the *functions* learned by the network. On the theory side, several works have appreciated the need to model the inputs, and to go beyond the simple component-wise i.i.d. modelling (Bruna & Mallat, 2013; Patel et al., 2016; Mézard, 2017; Gabrié et al., 2018; Mossel, 2018; Saxe et al., 2019). While we will focus on the ability of neural network to generalise from examples, two recent papers studied a network's ability to *store* inputs with lower-dimensional structure and random labels: Chung et al. (2018) studied the linear separability of general, finite-dimensional manifolds, while Rotondo et al. (2019) extended Cover's argument (Cover, 1965) to count the number of learnable dichotomies when inputs are grouped in tuples of $k$ inputs with the same label.

**Accessibility and reproducibility** The full code of our experiments can be accessed via `https://drive.google.com/open?id=1L0UOtOoRTYSHZtTxMxKIQuZLEuVaoJl_`. We give necessary parameter values to reproduce our figures beneath each plot. For ease of reading, we adopt the notation from the textbook by Goodfellow et al. (2016).

## 2 SETUP

In order to proceed on the question of what is a suitable model for structured data, we consider the setup of a feedforward neural network with one hidden layer with a few hidden units, as described below. We chose this setting because it is the simplest one we found where we were able to identify key differences between training in the vanilla teacher-student setup and training on the MNIST task. So throughout this work, we focus on the dynamics and performance of fully-connected two-layer neural networks with $K$ hidden units and first- and second-layer weights $\boldsymbol{W} \in \mathbb{R}^{K \times N}$ and $\boldsymbol{v} \in \mathbb{R}^K$, resp. Given an input $\mathbf{x} \in \mathbb{R}^N$, the output of a network with parameters $\boldsymbol{\theta} = (\boldsymbol{W}, \boldsymbol{v})$ is given by

$$\phi(\mathbf{x}; \boldsymbol{\theta}) = \sum_k^K v_k g\left(\boldsymbol{w}_k \mathbf{x}/\sqrt{N}\right), \tag{1}$$

where $\boldsymbol{w}_k$ is the $k$th row of $\boldsymbol{W}$, and $g : \mathbb{R} \to \mathbb{R}$ is the non-linear activation function of the network. We will focus on sigmoidal networks with $g(x) = \mathrm{erf}(x/\sqrt{2})$, or ReLU networks where $g(x) = \max(0, x)$ (see Appendix E).

We will train the neural networks on data sets with $P$ input-output pairs $(\mathbf{x}_i, y_i^*)$, $i = 1, \ldots, P$, where we use the starred $y_i^*$ to denote the *true* label of an input $\boldsymbol{x}_i$. We train networks by minimising the quadratic training error $E(\boldsymbol{\theta}) = 1/2 \sum_{i=1}^P \Delta_i^2$ with $\Delta_i = [\phi(\mathbf{x}_i, \boldsymbol{\theta}) - y_i^*]$ using stochastic gradient descent (SGD) with constant learning rate $\eta$,

$$\boldsymbol{\theta}^{\mu+1} = \boldsymbol{\theta}^\mu - \eta \nabla_{\boldsymbol{\theta}} E(\theta)|_{\boldsymbol{\theta}^\mu, \boldsymbol{x}_\mu, y_\mu^*}. \tag{2}$$

Initial weights for both layers of sigmoidal networks were always taken component-wise i.i.d. from the normal distribution with mean 0 and variance 1. The initial weights of ReLU networks were also taken from the normal distribution, but with variance $10^{-6}$ to ensure convergence.

The key quantity of interest is the *test error* or *generalisation error* of a network, for which we compare its predictions to the labels given in a test set that is composed of $P^*$ input-output pairs $(\mathbf{x}_i, y_i^*)$, $i = 1, \ldots, P^*$ that are *not* used during training,

$$\epsilon_g^{\mathrm{mse}}(\boldsymbol{\theta}) \equiv \frac{1}{2P^*} \sum_i^{P^*} [\phi(\mathbf{x}_i, \boldsymbol{\theta}) - y_i^*]^2. \tag{3}$$

The test set might be composed of MNIST test images or generated by the same probabilistic model that generated the training data. For binary classification tasks with $y^* = \pm 1$, this definition is easily amended to give the fractional generalisation error $\epsilon_g^{\mathrm{frac}}(\boldsymbol{\theta}) \propto \sum_i^{P^*} \Theta\left[-\phi(\mathbf{x}_i, \boldsymbol{\theta}) y_i^*\right]$, where $\Theta(\cdot)$ is the Heaviside step function.

### 2.1 LEARNING FROM REAL DATA OR FROM GENERATIVE MODELS?

We want to compare the behaviours of two-layer neural networks Eq. (1) trained either on real data sets or on unstructured tasks. As an example of a real data set, we will use the MNIST image database

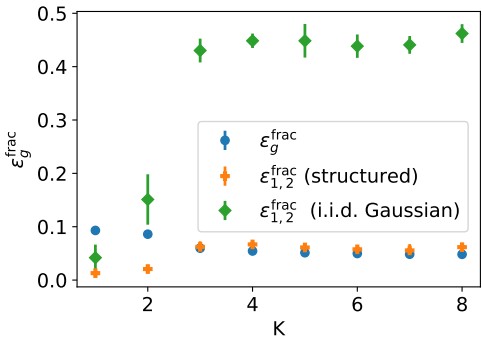 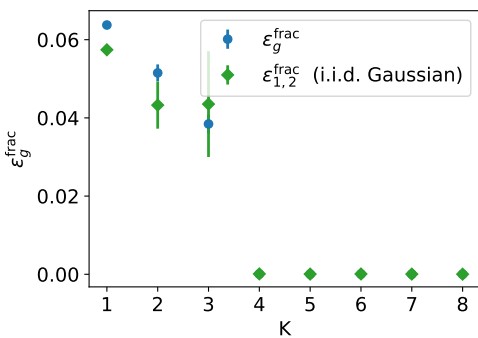

Figure 1: *(Left)* **Networks trained independently on MNIST achieve similar performance, but learn different functions.** For two networks trained independently on the MNIST odd-even classification task, we show the averaged final fractional test error, $\epsilon_g^{\text{frac}}$ (blue dots). We also plot $\epsilon_{1,2}^{\text{frac}}$ (5), the fraction of Gaussian i.i.d. inputs and MNIST test images the networks classify differently after training (green diamonds and orange crosses, resp.). *(Right)* **Training independent networks on a teacher task with i.i.d. inputs does not reproduce this behaviour.** We plot the results of the same experiment, but for Gaussian i.i.d. inputs with teacher labels $y_i^*$ (Eq. 4, $M = 4$). For both plots, $g(x) = \text{erf}\left(x/\sqrt{2}\right), \eta = 0.2, P = 76N, N = 784$.

of handwritten digits (LeCun & Cortes, 1998) and focus on the task of discriminating odd from even digits. Hence the inputs $\boldsymbol{x}_i$ will be the MNIST images with labels $y_i^* = 1, -1$ for odd and even digits, resp. The joint probability distribution of input-output pairs $(\boldsymbol{x}_i, y_i^*)$ for this task is inaccessible, which prevents analytical control over the test error and other quantities of interest. To make theoretical progress, it is therefore promising to study the generalisation ability of neural networks for data arising from a probabilistic generative model.

A classic model for data sets is the *vanilla teacher-student setup* (Gardner & Derrida, 1989), where unstructured i.i.d. inputs are fed through a random neural network called the *teacher*. We will take the teacher to have two layers and $M$ hidden nodes. We allow that $M \neq K$ and we will draw the components of the teacher's weights $\boldsymbol{\theta}^* = (\boldsymbol{v}^* \in \mathbb{R}^M, \boldsymbol{W}^* \in \mathbb{R}^{M \times N})$ i.i.d. from the normal distribution with mean zero and unit variance. Drawing the inputs i.i.d. from the standard normal distribution $\mathcal{N}(\mathbf{x}; 0, \boldsymbol{I}_N)$, we will take

$$y_i^* = \phi(\mathbf{x}_i, \boldsymbol{\theta}^*) \tag{4}$$

for regression tasks, or $y_i^* = \text{sgn}(\phi(\mathbf{x}_i, \boldsymbol{\theta}^*))$ for binary classification tasks. This is hence an example of a teacher task. In this setting, the network with $K$ hidden units that is trained using SGD Eq. (2) is traditionally called the *student*. Notice that, if $K \geq M$, there exists a student network that has zero generalisation error, the one with the same architecture and parameters as the teacher.

## 3 TWO CHARACTERISTIC BEHAVIOURS OF NEURAL NETWORKS TRAINED ON STRUCTURED DATA SETS

We now proceed to demonstrate experimentally two significant differences in the dynamics and the performance of neural networks trained on realistic data sets and networks trained within the vanilla teacher-student setup.

### 3.1 INDEPENDENT NETWORKS ACHIEVE SIMILAR PERFORMANCE, BUT LEARN DIFFERENT FUNCTIONS WHEN TRAINED ON STRUCTURED TASKS

We trained two sigmoidal networks with $K$ hidden units, starting from two independent draws of initial conditions to discriminate odd from even digits in the MNIST database. We trained both networks using SGD with constant learning rate $\eta$, eq. (2), until the generalisation error had converged to a stationary value. We plot this asymptotic fractional test error $\epsilon_g^{\text{frac}}$ as blue circles on the left

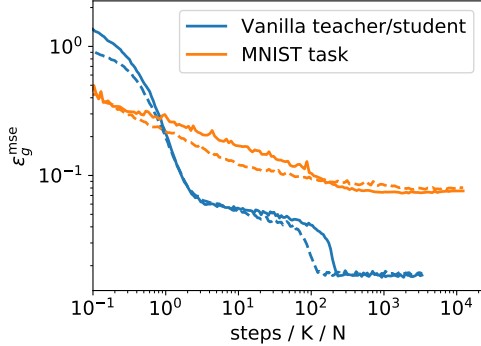 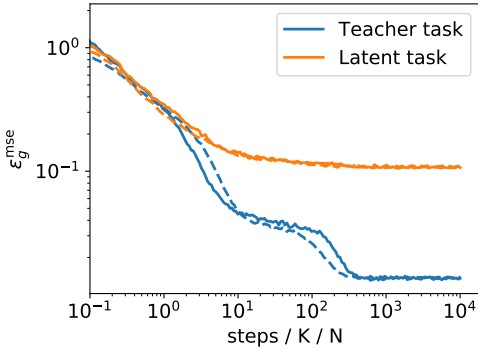

Figure 2: *(Left)* **Extended periods with stationary test error during training ("plateaus") appear in the vanilla teacher-student setup, not on MNIST.** We plot the generalisation error $\epsilon_g^{\mathrm{mse}}$ (3) of a network trained on Gaussian i.i.d. inputs with teacher labels (Eq. 4, $M=4$, blue) and when learning to discriminate odd from even digits in MNIST (orange). We trained either the first layer only (dashed) or both layers (solid). Notice the log scale on the x-axes. *(Right)* **Both structured inputs and latent labels are required to remove the plateau for synthetic data.** Same experiment, but now the network is trained on structured inputs (Eq. 6) ($f(x) = \mathrm{sgn}(x)$), with teacher labels $y_i^*$ (Eq. 4, blue) and with latent labels $\widetilde{y}_i^*$ (Eq. 7, orange). In both plots, $g(x) = \mathrm{erf}\left(x/\sqrt{2}\right), P = 76N, K = 3, \eta = 0.2$.

in Fig. 1 (the averages are taken over both networks and over several realisations of the initial conditions). We observed the same qualitative behaviour when we employed the early-stopping error to evaluate the networks, where we take the minimum of the generalisation error during training (see Appendix C).

First, we note that increasing the number of hidden units in the network decreases the test error on this task. We also compared the networks to one another by counting the fraction of inputs which the two networks classify differently,

$$\epsilon_{1,2}^{\mathrm{frac}}(\boldsymbol{\theta}_1, \boldsymbol{\theta}_2) \equiv \frac{1}{2P^*}\sum_i^{P^*} \Theta\left[-\phi(\mathbf{x}_i, \boldsymbol{\theta}_1)\phi(\mathbf{x}_i, \boldsymbol{\theta}_2)\right]. \tag{5}$$

This is a measure of the degree to which both networks have learned the same function $\phi(\mathbf{x}, \boldsymbol{\theta})$. Independent networks disagree on the classification of MNIST test images at a rate that roughly corresponds to their test error for $K \geq 3$ (orange crosses). However, even though the additional parameters of bigger networks are helpful in the discrimination task (decreasing $\epsilon_g$), both networks learn increasingly different functions when evaluated over the whole of $\mathbb{R}^N$ using Gaussian inputs as the network size $K$ increases (green diamonds). The network learned the right function on the lower-dimensional manifold on which MNIST inputs concentrate, but not outside of it.

This behaviour is not reproduced if we substitute the MNIST data set with a data set of the same size drawn from the vanilla teacher-student setup from Sec. 2.1 with $M = 4$, leaving everything else the same (right of Fig. 1). The final test error decreases with $K$, and as soon as the expressive power of the network is at least equal to that of the teacher, *i.e.* $K \geq M$, the asymptotic test error goes to zero, since the data set is large enough for the network to recover the teacher's weights to within a very small error, leading to a small generalisation error. We also computed the $\epsilon_{1,2}^{\mathrm{frac}}$ evaluated using Gaussian i.i.d. inputs (green diamonds). Networks with fewer parameters than the teacher find different approximations to that function, yielding finite values of $\epsilon_{1,2}$. If they have just enough parameters ($K = M$), they learn the same function. Remarkably, they also learn the same function when they have significantly *more* parameters than the teacher. The vanilla teacher-student setup is thus unable to reproduce the behaviour observed when training on MNIST.

## 3.2 THE GENERALISATION ERROR EXHIBITS PLATEAUS DURING TRAINING ON I.I.D. INPUTS

We plot the generalisation dynamics, *i.e.* the test error as a function of training time, for neural networks of the form (1) in Fig. 2. For a data set drawn from the vanilla teacher-student setup with $M = 4$, (blue lines in the left-hand plot of Fig. 2), we observe that there is an extended period of training during which the test error $\epsilon_g$ remains constant before a sudden drop. These "plateaus" are well-known in the literature for both SGD, where they appear as a function of time (Biehl & Schwarze, 1995; Saad & Solla, 1995; Biehl et al., 1996), and in batch learning, where they appear as a function of the training set size (Schwarze, 1993; Engel & Van den Broeck, 2001). Their appearance is related to different stages of learning: After a brief exponential decay of the test error at the start of training, the network "believes" that data are linearly separable and all her hidden units have roughly the same overlap with all the teacher nodes. Only after a longer time, the network picks up the additional structure of the teacher and "specialises": each of its hidden units ideally becomes strongly correlated with one and only one hidden unit of the teacher before the generalisation error decreases exponentially to its final value.

In contrast, the generalisation dynamics of the same network trained on the MNIST task (orange trajectories on the left of Fig. 2) shows no plateau. In fact, plateaus are rarely seen during the training of neural networks (note that during training, we do not change any of the hyper-parameters, *e.g.* the learning rate $\eta$.)

It has been an open question how to eliminate the plateaus from the dynamics of neural networks trained in the teacher-student setup. The use of second-order gradient descent methods such as natural gradient descent (Yang & Amari, 1998) can shorten the plateau (Rattray et al., 1998), but we would like to focus on the more practically relevant case of first-order SGD. Yoshida et al. (2019) recently showed that length and existence of the plateau depend on the dimensionality of the output of the network, but we would like a model where the plateau disappears independently of the output dimension.

## 4 THE HIDDEN MANIFOLD MODEL

We now introduce a new generative probabilistic model for structured data sets with the aim of reproducing the behaviour observed during training on MNIST, but with a synthetic data set. The main motivation for such a model is that a closed-form solution of the learning dynamics is expected to be accessible. To generate a data set containing $P$ inputs in $N$ dimensions, we first choose $D$ feature vectors in $N$ dimensions and collect them in a feature matrix $\mathbf{F} \in \mathbb{R}^{D \times N}$. Next we draw $P$ vectors $\mathbf{c}_i$ with random i.i.d. components and collect them in the matrix $\mathbf{C} \in \mathbb{R}^{P \times D}$. The vector $\mathbf{c}_i$ gives the coordinates of the $i$th input on the lower-dimensional manifold spanned by the feature vectors in $\mathbf{F}$. We will call $\mathbf{c}_i$ the *latent representation* of the input $\mathbf{x}_i$, which is given by the $i$th row of

$$\boldsymbol{X} = f\left(\mathbf{CF}/\sqrt{D}\right) \in \mathbb{R}^{P \times N}, \tag{6}$$

where $f$ is a non-linear function acting component-wise. In this model, the "world" of the data on which the true label can depend is a $D$-dimensional manifold, which is obtained from the linear subspace of $\mathbb{R}^N$ generated by the $D$ lines of matrix $\mathbf{F}$, through a folding process induced by the nonlinear function $f$. As we discuss in Appendix A, the exact form of $f$ does not seem to be important, as long as it is a nonlinear function.

The latent labels are obtained by applying a two-layer neural network with weights $\tilde{\boldsymbol{\theta}}^* = (\tilde{\mathbf{W}}^* \in \mathbb{R}^{M \times D}, \tilde{\mathbf{v}}^* \in \mathbb{R}^M)$ within the unfolded hidden manifold according to

$$\widetilde{y}_i^* = \phi(\mathbf{c}_i, \widetilde{\boldsymbol{\theta}}^*) = \sum_m^M \tilde{v}_m^* g\left(\widetilde{\mathbf{w}}_m^* \mathbf{c}_i/\sqrt{D}\right). \tag{7}$$

We draw the weights in both layers component-wise i.i.d. from the normal distribution with unity variance, unless we note it otherwise. The key point here is the dependency of labels $\widetilde{y}_i$ on the coordinates of the lower-dimensional manifold $\mathbf{C}$ rather than on the high-dimensional data $\mathbf{X}$. We believe that the exact form of this dependence is not crucial and we expect several other choices to yield similar results to the ones we will present in the next section.

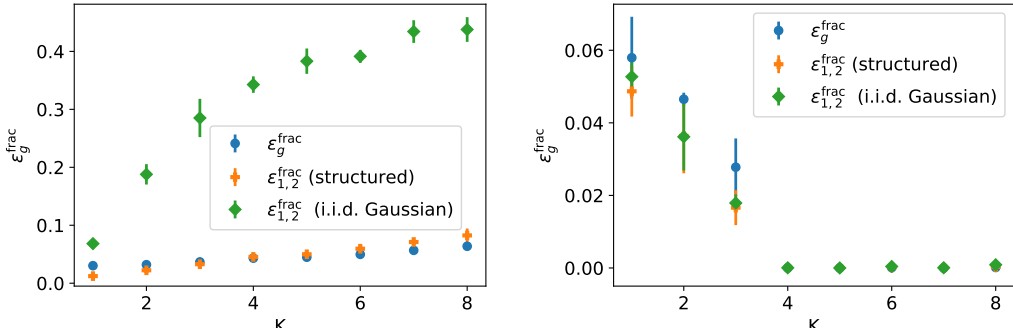

Figure 3: **A latent task on structured inputs makes independent networks behave like networks trained on MNIST.** *(Left)* For two networks trained independently on a binary classification task with structured inputs (6) and latent labels $\widetilde{y}_i^*$ (Eq. 7, $M = 1$), we plot the final fractional test error, $\epsilon_g^{\text{frac}}$ (blue dots). We also plot $\epsilon_{1,2}^{\text{frac}}$ (5), the fraction of Gaussian i.i.d. inputs and structured inputs the networks classify differently after training (green diamonds and orange crosses, resp.). *(Right)* In the same experiment, structured inputs with *teacher labels* $y_i^*$ (4) ($M = 4$) fail to reproduce the behaviour observed on MNIST (cf. Fig. 1). In both plots, $f(x) = \text{sgn}(x), g(x) = \text{erf}\left(x/\sqrt{2}\right), D = 10, \eta = 0.2$.

In the following, we choose the entries of both $\mathbf{C}$ and $\mathbf{F}$ to be i.i.d. draws from the normal distribution with mean zero and unit variance. To ensure comparability of the data sets for different data-generating function $f(x)$, we always center the input matrix $\boldsymbol{X}$ by subtracting the mean value of the entire matrix from all components and we rescale inputs by dividing all entries by the covariance of all the entries in the matrix before training.

## 4.1 THE IMPACT OF THE HIDDEN MANIFOLD MODEL ON NEURAL NETWORKS

We repeated the experiments with two independent networks reported in Sec. 3.1 using data sets generated from the hidden manifold model with $D = 10$ latent dimensions (see Appendix D). On the right of Fig. 3, we plot the asymptotic performance of a network trained on structured inputs which lie on a manifold (6) with a teacher task: the labels are a function of the high-dimensional inputs and do not explicitly take the latent representation $\mathbf{c}_i$ of an input into account, $y_i^* = \phi(\mathbf{x}_i, \boldsymbol{\theta}^*)$. The final results are similar to those of networks trained on data from the vanilla teacher-student setup (*cf.* right of Fig. 1): given enough data, the network recovers the teacher function if the network has at least as many parameters as the teacher. Once the teacher weights are recovered by both networks, they achieve zero test error (blue circles) and they agree on the classification of random Gaussian inputs because they do implement the same function.

The left plot of Fig. 3 shows network performance when trained on the same inputs, but this time with a latent task where the labels are a function of the latent representation of the inputs: $\widetilde{y}_i = \phi(\mathbf{c}_i, \widetilde{\boldsymbol{\theta}}^*)$. The asymptotic performance of the networks then resembles that of networks trained on MNIST: after convergence, the two networks will disagree on structured inputs at a rate that is roughly their generalisation error, but as $K$ increases, they also learn increasingly different functions, up to the point where they will agree on their classification of a random Gaussian input in just half the cases. The hidden manifold model thus reproduces the behaviour of independent networks trained on MNIST.

A look at the right-hand plot Fig. 2 reveals that in this model the plateaus are absent. Again, we repeat the experiment of Sec. 3.2, but we train networks on structured inputs $\boldsymbol{X} = \text{sgn}(\mathbf{CF})$ with teacher $(y_i^*)$ and latent labels $(\widetilde{y}_i^*)$, respectively. It is clear from these plots that the plateaus only appear for the teacher task. In Appendix B, we demonstrate that the lack of plateaus for latent tasks in Fig. 2 is *not* due to the fact that the network in the latent task asymptotes at a higher generalisation error than the teacher task.

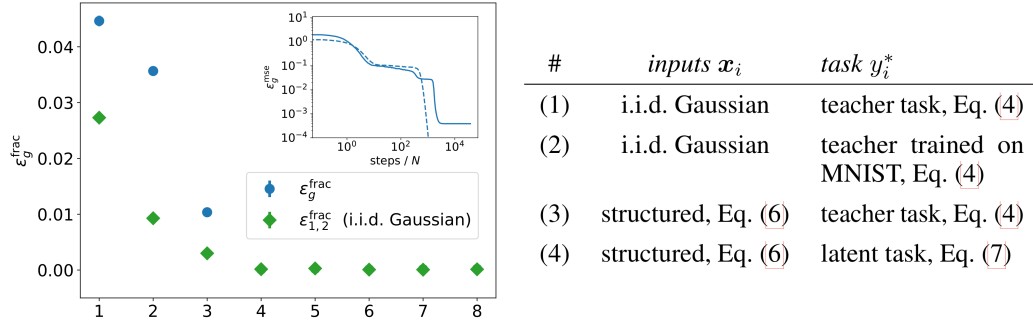

Figure 4: *(Left)* Same plot as the right plot of Fig. 1 with Gaussian i.i.d. inputs $\boldsymbol{x}_i$ and labels $y_i^*$ (4) provided by a teacher network with $M = 4$ hidden units that was pre-trained on the MNIST task, reaching $\sim 5\%$ on the task. *Inset:* Typical generalisation dynamics of networks where we train the first or both layers (dashed and solid, resp.). $g(x) = \text{erf}\left(x/\sqrt{2}\right), \eta = 0.2, N = 784, M = K = 4, P = 76N$. *(Right)* Four different setups for synthetic data sets in supervised learning problems.

## 4.2 LATENT TASKS, STRUCTURED INPUTS ARE BOTH NECESSARY TO MODEL REAL DATA SETS

Our quest to reproduce the behaviour of networks trained on MNIST has led us to consider three different setups so far: the vanilla teacher-student setup, *i.e.* a teacher task on unstructured inputs; and teacher and latent tasks on structured inputs. While it is not strictly possible to test the case of a latent task with unstructured inputs, we can approximate this setup by training a network on the MNIST task and then using the resulting network as a teacher to generate labels $y_i^*$ (4) for inputs drawn i.i.d. component-wise from the standard normal distribution. To test this idea, we trained both layers sigmoidal networks with $M = 4$ hidden units using vanilla SGD on the MNIST task, where they reach a generalisation error of about $5\%$. They have thus clearly learnt some of the structure of the MNIST task. However, as we show on the left of Fig. 4, independent students trained on a data set with i.i.d. Gaussian inputs $\boldsymbol{x}_i$ and true labels $y_i^*$ given by the pre-trained teacher network behave similarly to students trained in the vanilla teacher-student setup of Sec. 3.1. Furthermore, the learning dynamics of a network trained in this setup display the plateaus that we observed in the vanilla teacher-student setup (inset of Fig. 4).

On the right of Fig. 4, we summarise the four different setups for synthetic data sets in supervised learning problems that we have analysed in this paper. Only the hidden manifold model, consisting of a latent task on structured inputs, reproduced the behaviour of neural networks trained on the MNIST task, leading us to conclude that a model for realistic data sets has to feature both, structured inputs and a latent task.

## 5 CONCLUDING PERSPECTIVES

We have introduced the hidden manifold model for structured data sets that is simple to write down, yet displays some of the phenomena that we observe when training neural networks on real-world inputs. We saw that the model has two key ingredients, both of which are necessary: (1) high-dimensional inputs which lie on a lower-dimensional manifold and (2) latent labels for these inputs that depend on the inputs' position within the low dimensional manifold. We hope that this model is a step towards a more thorough understanding of how the structure we find in real-world data sets impacts the training dynamics of neural networks and their ability to generalise.

We see two main lines for future work. On the one hand, the present work needs to be generalised to multi-layer networks to identify how depth helps to deal with structured data sets and to build a model capturing the key properties. On the other hand, the key promise of the synthetic hidden manifold model is that the learning dynamics should be amenable to closed-form analysis in some limit. Such analysis and its results would then provide further insights about the properties of learning beyond what is possible with numerical experiments.

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

# A  THE EXACT FORM OF THE DATA-GENERATING FUNCTION $f(\cdot)$ IS NOT IMPORTANT, AS LONG AS IT IS NON-LINEAR

Two questions arise when looking at the way we generate inputs in our data sets, $\boldsymbol{X} = f\left(\mathbf{CF}/\sqrt{D}\right)$: is the non-linearity necessary $f(\cdot)$ necessary, and is the choice of non-linearity important?

To answer the first question, we plot the results of the experiment with independent networks described in Sec. 4.1. The setup is exactly the same, except that we now take inputs to be

$$\boldsymbol{X} = \mathbf{CF}, \tag{8}$$

i.e. inputs are just a linear combination of the feature vectors, without applying a non-linearity. In this case, two networks trained in the vanilla teacher-student setup will learn globally different functions, as can be seen from the fractional generalisation error between the networks (5) (green diamonds), which is $1/2$, i.e. not better than chance. This is a direct consequence of using $f(x) = x$: to perfectly generalise with respect to the teacher, it is thus sufficient to learn only the $D$ components of the teacher weights $\boldsymbol{w}_m^*$ in the direction $\mathbf{F}$. Thus the weights of the network in the weight space orthogonal to the directions $\mathbf{F}$ are unconstrained, and by starting from random initial conditions, will converge to different values for each network.

We also checked that the qualitative behaviour of a neural networks trained on the hidden manifold model does not depend on the data-generating non-linearity $f(x)$. In Fig. 6, we therefore show the results of the same experiment described in Sec. 4.1, but this time using

$$\boldsymbol{X} = \max\left(0, \mathbf{CF}\right). \tag{9}$$

where the application of the non-linearity is again component-wise. Indeed, the results mirror those when we used the sign function $f(x) = \operatorname{sgn}(x)$.

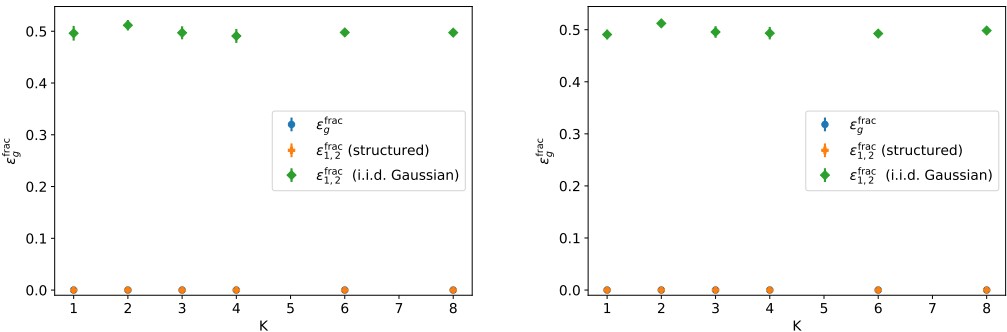

Figure 5: **The input-generating function must be non-linear.** We repeat the plots of Fig. 3, where we plot the fractional test errors of networks trained on labels generated by a teacher with $M = 1$ hidden units acting on the inputs *(Left)* and on the coefficients *(Right)*, only that we take the inputs to be $\boldsymbol{X} = \mathbf{CF}$, i.e. we choose a linear data-generating function $f(x) = x$. Notably, even networks trained within the vanilla teacher-student setup will disagree on Gaussian inputs. $M = 1, \eta = 0.2, D = 10, \tilde{v}_m^* = 1$.

# B  THE EXISTENCE OF PLATEAUS DOES NOT DEPEND ON THE ASYMPTOTIC GENERALISATION ERROR

We have demonstrated on the right of Fig. 2 that neural networks trained on data drawn from the hidden manifold model (HMF) introduced here do not show the plateau phenomenon, where the generalisation error stays stationary after an initial exponential decay, before dropping again. Upon closer inspection, one might think that this is due to the fact that the student trained on data from the HMF asymptotes at a higher generalisation error than the student trained in the vanilla teacher-student setup. This is not the case, as we demonstrate in Fig. 7: we observe no plateau in a sigmoidal network

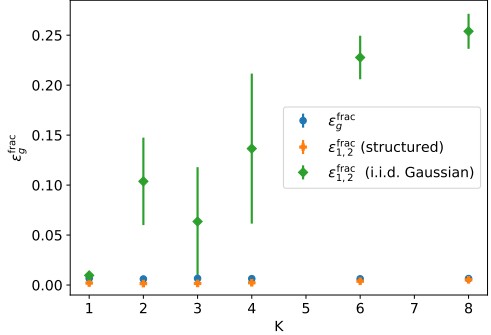
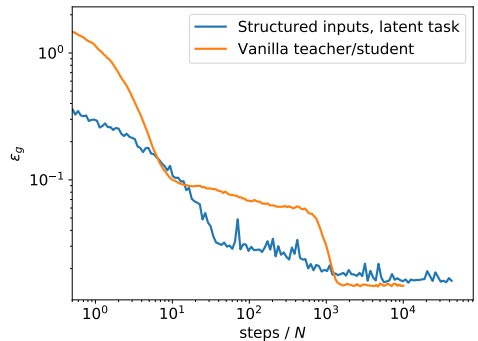

Figure 6: **The qualitative behaviour of independent students trained on the hidden manifold model does not depend on our choice of data-generating non-linearity** $f(x)$. Same plot as Fig. 3, with $\boldsymbol{X} = \max(0, \boldsymbol{CF})$. $M = 1, \eta = 0.2, D = 10, \tilde{v}_m^* = 1$.

Figure 7: **The plateau in the vanilla teacher-student setup can have larger generalisation error than the asymptotic error in a latent task on structured inputs.** Generalisation dynamics of a sigmoidal network where we train only the first layer on (i) structured inputs $\boldsymbol{X} = \max(0, \boldsymbol{CF})$ with latent labels $\tilde{y}_i$ (7) (*blue*, $D = 10$) and (ii) the vanilla teacher-student setup (Sec. 2, *orange*). In both cases, $M = 5, K = 6, \eta = 0.2, P = 76N, v_m^* = v^* = 1$.

trained on data from the HMF even that network asymptotes at a generalisation error that is, within fluctuations, the same as the generalisation error of a network of the same sized trained in the vanilla teacher-student setup and which shows a plateau.

## C EARLY-STOPPING YIELDS QUALITATIVELY SIMILAR RESULTS

In Fig. 8, we reproduce Fig. 3, where we compare the performance of independent neural networks trained on the MNIST task *(Left)*, or trained on structured inputs with a latent task *(Center)* and a teacher task *(Right)*, respectively. This time, we the early-stopping generalisation error $\hat{\epsilon}_g^{\text{frac}}$ rather than the asymptotic value at the end of training. We define $\hat{\epsilon}_g^{\text{frac}}$ as the minimum of $\epsilon_g^{\text{frac}}$ during the whole of training. Clearly, the qualitative result of Sec. 4.1 is unchanged: although we use structured inputs (6) in both cases, independent students will learn different functions which agree on those inputs only when they are trained on a latent task (7) *(Center)*, but not when trained on a vanilla teacher task (4) *(Right)*. Thus structured inputs and latent tasks are sufficient to reproduce the behaviour observed when training on the MNIST task.

## D DYNAMICS WITH A LARGE NUMBER OF FEATURES $D \sim N$

It is of independent interest to investigate the behaviour of networks trained on data from the hidden manifold model when the number of feature vectors $D$ is on the same order as the input dimension $N$. We call this the regime of extensive $D$. It is a different regime from MNIST, where experimental studies consistently find that inputs lie on a low-dimensional manifold of dimension $D \sim 14$, which is much smaller than the input dimension $N = 784$ (Costa & Hero, 2004; Levina & Bickel, 2004; Spigler et al., 2019).

We show the results of our numerical experiments with $N = 500, D = 250$ in Fig. 9, where we reproduce Fig. 3 for the asymptotic (top row) and the early-stopping (bottow row) generalisation error. The behaviour of networks trained on a teacher task with structured inputs (right column) is unchanged w.r.t. to the case with $D = 10$. For the latent task, increasing the number of hidden units however *increases* the generalisation error, indicating severe over-fitting, which is only partly mitigated by early stopping. The generalisation error on this task is generally much higher than in the low-$D$ regime and clearly, increasing the width of the network is not the right way to learn a latent

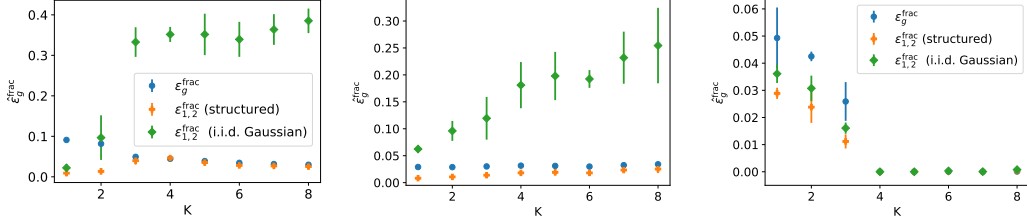

Figure 8: **Measuring early stopping errors does not affect the phenomenology of latent and teacher tasks.** *(Left)* Performance of independent sigmoidal students on the MNIST task as evaluated by the early-stopping generalisation error. *(Center* and *Right)* We reproduce Fig. 3 of the main text, but this time we plot the early-stopping generalisation error $\hat{\epsilon}_g^{\text{frac}}$ for two networks trained independently on a binary classification task with structured inputs (6) and latent labels $\widetilde{y}_i^*$ (Eq. 7, $M = 1$, *Center*) and teacher labels $y_i^*$ (4) ($M = 4$) *(Left)*. In both plots, $f(x) = \text{sgn}(x), g(x) = \text{erf}\left(x/\sqrt{2}\right), D = 10, \eta = 0.2$.

task; instead, it would be intriguing to analyse the performance of deeper networks on this task where finding a good intermediate representation for inputs is key. This is an intriguing avenue for future research.

# E  INDEPENDENT STUDENTS WITH RELU ACTIVATION FUNCTION

We also verified that the behaviour of independent networks we observed on MNIST with sigmoidal students persists when training networks with ReLU activation function and that the hidden manifold model is able to reproduce it for these networks. We show the results of our numerical experiments in Fig. 10. To that end, we trained both layers of a network $\phi(\boldsymbol{x}, \boldsymbol{\theta})$ with $g(x) = \max(x, 0)$ starting from small initial conditions, where we draw the weights component-wise i.i.d. from a normal distribution with variance $10^{-6}$.

We see that the generalisation error of ReLU networks on the MNIST task *(Left* of Fig. 10) decreases with increasing number of hidden units, while the generalisation error on MNIST inputs of the two independent students with respect to each other is comparable or less than the generalisation error of each individual network on the MNIST task.

On structured inputs with a teacher task *(Right* of Fig. 10), where labels were generated by a teacher with $M = 4$ hidden units, the student recovers the teacher such that its generalisation error is less than $10^{-3}$ for $K > 4$, and both independent students learn the same function, as evidenced by their generalisation errors with respect to each other. This is the same behaviour that we see in Fig. 3 for sigmoidal networks. The finite value of the generalisation error for $K = M = 4$ is due to two out of ten runs taking a very long time to converge, longer than our simulation lasted for. Finally, we see that for a latent task on structured inputs, the generalisation error of the two networks with respect to each other increases beyond the generalisation error on structured inputs of each of them, as we observed on MNIST. Thus we have recovered the phenomenology that we described for sigmoidal networks in ReLU networks, too.

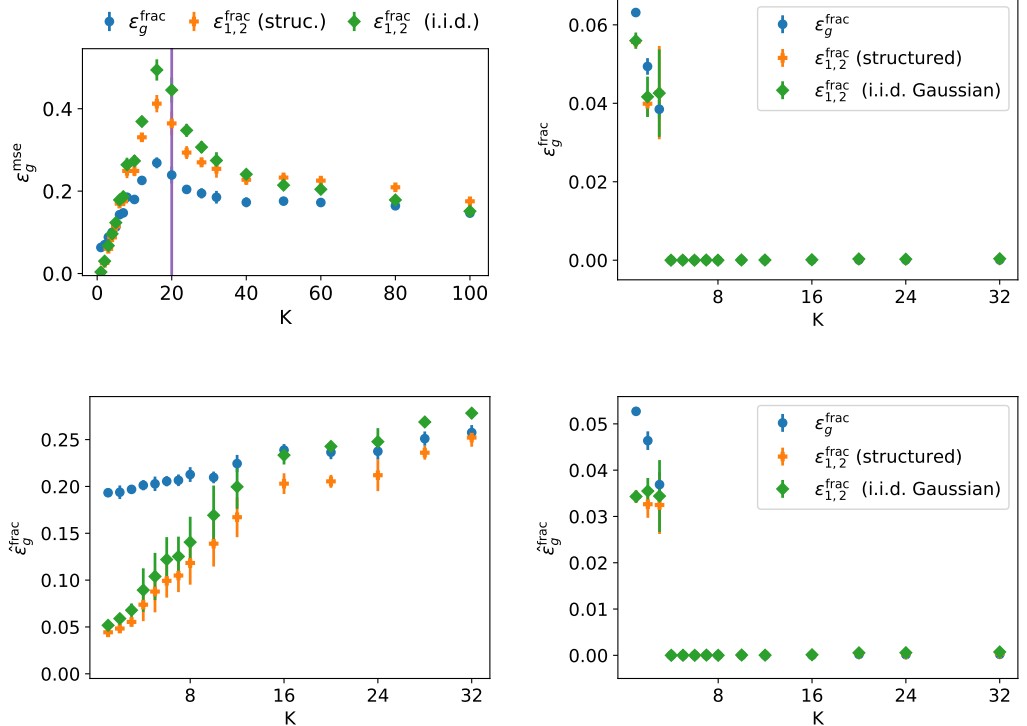

Figure 9: **Performance of independent networks trained on a latent task with inputs in many latent directions** $D = N/2$. *(Top Left)* For two networks trained independently on a binary classification task with structured inputs (6) and latent labels $\widetilde{y}_i^*$ (Eq. 7, $M = 1$), we plot the final fractional test error, $\epsilon_g^{\mathrm{frac}}$ (blue dots). We also plot $\epsilon_{1,2}^{\mathrm{frac}}$ (5), the fraction of Gaussian i.i.d. inputs and structured inputs the networks classify differently after training (green diamonds and orange crosses, resp.). *(Top Right)* Same experiment, but with structured inputs and *teacher labels* $y_i^*$ (4) ($M = 4$). *(Bottom row)* Same plots as in the top row, but this time for the early-stopping error $\hat{\epsilon}^{\mathrm{frac}}$ (see Sec. C). In all plots, $f(x) = \mathrm{sgn}(x), g(x) = \mathrm{erf}\left(x/\sqrt{2}\right), N = 500, D = 250, \eta = 0.2$.

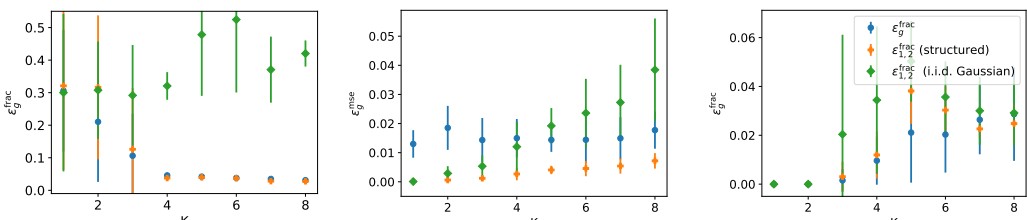

Figure 10: **Behaviour of independent students with ReLU activation functions.** *(Left)* Asymptotic generalisation error of independent students with ReLU activation function $g(x) = \max(0, x)$ on the MNIST task. *(Center* and *Right)* We reproduce Fig. 3 of the main text for two networks with ReLU activation trained independently on a binary classification task with structured inputs (6) and latent labels $\widetilde{y}_i^*$ (Eq. 7, $M = 1$) *(Center)* and teacher labels $y_i^*$ (4) ($M = 4$ *Right*). In both plots, $f(x) = \mathrm{sgn}(x), g(x) = \max(0, x), D = 10, \eta = 0.1$.

