# OpenReview forum: "Modelling the influence of data structure on learning in neural networks"
_ICLR.cc/2020/Conference — Reject_

### Official Review · AnonReviewer3 · 2019-10-23
**Official Blind Review #3**

**Rating:** 1

**Review:**

The paper studies how different settings of data structure affect learning of neural networks and how to mimic the behavior of neural networks seen on real datasets (e.g. MNIST) when learning on a synthetic one.
I would recommend rejecting the paper due to several issues pointed out below.
1. The paper abuses blanket citation making it difficult to identify and verify contribution and conduct a comparison with existing literature. Related work section amounts only to one paragraph in size. It looks questionable, that nobody ever treated the problem of the generalization ability of neural networks from a point of the data manifold properties. After a quick search, for example, [1] provides an in-depth analysis and generalization bounds for two-layered neural networks and provides a data complexity measure that can discriminate between random labels (which are equivalent to the outputs of randomly initialized fixed teacher networks in this work) and true labels on structured datasets like MNIST and CIFAR. The paper fails to cite this work as well.
2. The paper claims to experimentally identify key differences in the training dynamics of neural networks in teacher-student setup and on an MNIST task (binary classification into even and odd numbers).  One of the differences is the presence of plateaus in the learning curves in the vanilla teacher-student setup, however, as the paper states itself - this is a well established and studied characteristic of the setup, not something unexpected and new.
3. Overall, I have yet to see actionable development in this paper as it consists of observations that have been noticed and studied previously and presents no attempt at explanation or rigorous analysis.

As for the experiments:
The setting of the experiment in section 3.1 leaves space for improvement. For example, it would be interesting to see whether two neural networks learned in the vanilla teacher-student setup on the iid random inputs agree on the MNIST inputs (i.e. inverting the experiment in 3.1) as a sanity check for other factors interplay since MNIST inputs would be an example of the out of distribution inputs for the networks learned on iid random examples used in the experiment. As another pointer, [2] shows that even when training input is from a standard normal distribution, the problem can have spurious local minima, implying that even on unstructured training datasets, neural networks from different initializations yield diverse outputs for out of distribution inputs, not agreeing among each other.

I would also like to see concrete examples when and how the hidden manifold model may benefit theoretical understanding or practical knowledge on, for example, how to cook a dataset or check if the dataset admits/affects learning.

References
[1] Arora, Sanjeev, et al. Fine-grained analysis of optimization and generalization for overparameterized two-layer neural networks.
[2] Safran, Itay, and Ohad Shamir. Spurious local minima are common in two-layer relu neural networks.

**Experience Assessment:**

I have read many papers in this area.

**Review Assessment: Checking Correctness Of Derivations And Theory:**

I carefully checked the derivations and theory.

**Review Assessment: Checking Correctness Of Experiments:**

I carefully checked the experiments.

**Review Assessment: Thoroughness In Paper Reading:**

I read the paper at least twice and used my best judgement in assessing the paper.

---

> ### Author Response · Authors · 2019-11-14
> **Our response**
>
> We thank the reviewer for his/her comments on our paper.
>
> 1. We are not sure what the referee means by a "blanket citation"? Can the referee be more concrete? We cite related work in the whole introduction, amounting to [33] papers. In the paragraph titled "related work"
> we merely summarised those that did not otherwise naturally fit into the introduction. We renamed the Introduction and that mentioned paragraph to avoid giving the wrong impression.
>
> The only specific comment about our "abuse of citations" is the reference to work [1]. While this work is interesting, it has respectfully no significant connection to our work. We do not study generalizations bounds, nor over-parametrization in our paper.
>
> 2. We did not mean to claim that identification of this difference if novel to our paper, indeed as the referee notes we say the opposite. But we see how our presentation might have been misleading and we adjusted the wording to make it clear that the identified difference is our method, while the main contribution is the model that reproduces them.
>
> 3. In most of natural science having a simple synthetic model reproducing the observed behaviour (or more of the observed behaviour than previous models) is seen as important progress even when it comes with no theorems and even when it does not explain ALL possible observed phenomena.
>
>
> Experiments:
>
> We thank the referee for the comment concerning the suggested test with MNIST inputs.
> Since the reviews lead overall to the rejection of our paper, we will test this thoroughly later. But we strongly expect to find that the two independently trained students do agree on MNIST inputs, contrary to what the referee suggest (if we understood his point correctly). Our point is not that when you train on one distribution you will disagree on another (as in our understanding suggested by the referee), but that when you train on a low-dimensional manifold inputs then you disagree outside of the manifold. The vanilla teacher student model inputs are not on low-dimensional manifold and hence two independent students do agree, even on MNIST inputs.
>
> The pointer related to paper [2] does not seem relevant. The fact that the vanila teacher-student model has (or can have) spurious local minima does not imply that two randomly initialized students will yield diverse outputs. We show that on the vanilla teacher student two independently trained students will with high probability lead to agreement comparable to the generalization accuracy. I.e. we will not see the behaviour of good generalization and large disagreement because of existence of several spurious local minima. With exponentially rare initialization we could, but this is not what we test in our paper, where the system sizes are large enough that the results we observe basically concentrate from one realization to another. These concentration properties are well studied and proven in the large size limit for the vanilla teacher-student models.
>
> The comment about theoretical understanding of the model is very fair and we agree. This is something we are working on and the 10 days we have for the revision are not enough to consolidate and report our theoretical findings.

---

### Official Review · AnonReviewer2 · 2019-10-25
**Official Blind Review #2**

**Rating:** 1

**Review:**

This paper studies influences of data structures on neural network learning. The data structures discussed in this paper are structured inputs (concentrating on a low-dimensional manifold) versus unstructured ones, as well as the teacher task (labels are obtained as a function of high-dimensional inputs) versus the latent task (labels are obtained as a function of the lower-dimensional latent representation of inputs). The introduced model, the hidden manifold model, which is a latent task with structured inputs, is claimed to reproduce two features found in learning of the MNIST data set, whereas the teacher task with unstructured inputs does not.

The observation that typical real-world datasets are concentrated on a lower-dimensional manifold is not novel, and it is also well expected that networks trained with such a dataset would exhibit different behaviors for inputs outside such a lower-dimensional manifold. The other observation that in real-world learning tasks one rarely encounters plateaux is not novel either. The possible novelty of this paper would thus be in the proposal of the hidden manifold model, but I am not convinced with the significance of the latent task. Because of these, I would judge possible contributions of this paper rather weak, so that I would not be able to recommend acceptance of this paper.

I think that the main difference between the authors’ “teacher task” and “latent task” lies in realizability of the underlying function: The teacher task is certainly realizable once the number of hidden units exceeds that of the teacher, whereas we are not sure about the realizability of the latent tasks. There might even be different levels of unrealizability which can affect learning. Anyway, the teacher versus latent distinction of learning tasks, as introduced in this paper, should be best regarded, at least in its current status, as a working hypothesis which would need more investigation. I would agree that this paper puts a step forward, but does not arrive at any decisive conclusion yet.

Page 3, line 26: The assumption that g should act componentwise does not seem needed because in equation (1) it acts on a scalar.
Page 4, line 16: there exist(s) a student network
Page 6, line 20: gradient descent methods such (as) natural gradient descent
Page 7, line 7: I do not understand what is meant by “by dividing all entries by the covariance of the entire matrix”. An entry should be a real number, whereas the covariance should be a matrix.
Page 7, line 12: cf. (left -> right) of Fig. 1


**Experience Assessment:**

I have read many papers in this area.

**Review Assessment: Checking Correctness Of Derivations And Theory:**

I assessed the sensibility of the derivations and theory.

**Review Assessment: Checking Correctness Of Experiments:**

I assessed the sensibility of the experiments.

**Review Assessment: Thoroughness In Paper Reading:**

I read the paper at least twice and used my best judgement in assessing the paper.

---

> ### Author Response · Authors · 2019-11-14
> **Our response**
>
> We thank the reviewer for his/her comments on our paper.
>
> We did not mean to claim that the observation that data lie on lower-dimensional manifold is novel, we indeed cite work [7, 8, 9, 10] where this appeared. The same for the absence of plateau is not novel. We see how our "main contribution" paragraph might have lead to this misunderstanding and we changed it accordingly in the revised manuscript. Our main contribution is indeed the simple synthetic model reproducing these observations.
>
> The referee states that he/she is not convinced by the significance of the latent task without giving any reason for this. Can the referee please express his/her reason for this so that we get a chance to address this criticism?
>
> The referee says that according to him/her the main difference is in the realizability or not of the task. This is certainly a point worth studying more thoroughly. We have not seen the MNIST properties in the basic ways un-realizability is usually put in the teacher-student model. But showing that what we report cannot be explained via realizability is again not the point of the paper and would be a topic for another paper.
>
> We do thank the reviewer to the list of 5 minor issues that we corrected.

---

> > ### Comment · AnonReviewer2 · 2019-11-14
> > **Clarification**
> >
> > > The referee states that he/she is not convinced by the significance of the latent task without giving any reason for this. Can the referee please express his/her reason for this so that we get a chance to address this criticism?
> >
> > Just for clarification, my reasoning was described in the next paragraph of the review comment.

---

> > > ### Author Response · Authors · 2019-11-15
> > > **Thank you for the clarification !**
> > >
> > > We commented on the issue of realisability that you raised in that paragraph in our next paragraph!

---

### Official Review · AnonReviewer1 · 2019-10-25
**Official Blind Review #1**

**Rating:** 3

**Review:**

The authors consider the general problem of "structure" in datasets--particularly, what are the features of datasets that govern the learning dynamics of neural networks trained to classify that data.  They approach this problem by looking at combinations of [iid gaussian, structure] inputs and [teacher, latent] tasks (for particular choices of "teacher", "latent", and "structure).  Finally, they identify that "structure" in the input space, and a notion of "latent"-ness in the task seem crucial for a synthetic dataset to recapitulate the learning dynamics of a real-world dataset.

The experiments, exposition, and motivation are all exceedingly clear.  My only reservations are about the scope of the experimentation / strength of conclusions of the paper for generally structure data.

Thus, I suggest a Weak Reject of this paper (though I would likely increase my rating to Weak Accept given my comments below).

The primary weakness of this paper is the over-reliance on the MNIST dataset, which is very nearly linearly separable.  Thus, I strongly worry that any notions of latentness that work for MNIST might not transfer at all to more complicated data regimes---i.e., while I believe the authors have identified and patched an interesting gap between the learning dynamics of iid data and MNIST, I'm not sure if there still isn't a gap between something like MNIST and, say, CIFAR-10.  I would raise my score to an accept if the authors carried out their analysis on (at least) CIFAR-10 as well, and even higher if the authors greatly expanded their experiments.

**Experience Assessment:**

I have read many papers in this area.

**Review Assessment: Checking Correctness Of Derivations And Theory:**

I assessed the sensibility of the derivations and theory.

**Review Assessment: Checking Correctness Of Experiments:**

I assessed the sensibility of the experiments.

**Review Assessment: Thoroughness In Paper Reading:**

I read the paper thoroughly.

---

> ### Author Response · Authors · 2019-11-14
> **Our response**
>
> We thank the reviewer for appreciating our work and summarising well what we aimed to do.
>
> Concerning our reliance on MNIST: Out point is that there is already a noticeable difference between the behaviour of the canonical teacher-student model and learning with MNIST. In more complex datasets such as CIFAR there will only be more differences. Our main point is to construct a model that will reproduce the behaviour observe in MNIST better than the canonical teacher-student model.
>
> We are not sure that going further and identifying key differences between learning on MNIST and, say, CIFAR is an interesting next step. While the need of convolutional layer will almost surely play against analytical tractability of an eventual corresponding model, the role of depth is for instance a direction that we highlighted as important. But this does goes beyond the scope of the present paper.

---

> > ### Comment · AnonReviewer1 · 2019-11-15
> > **Response to response**
> >
> > I suppose my point was more of a meta-scientific one: many projects aim to identify something interesting by comparing between [synthetic toy dataset] and MNIST, and many such projects find something interesting.  The moment those results are tried on [something more complicated than MNIST], they either fail, or the story becomes so muddy as to challenge the validity of the originally "interesting thing".
> >
> > If I can compress my criticism to a sentence: It's difficult to evaluate the generality of a method when it's restricted only to a single comparison two datasets.  I do not mean to discourage the authors--I think this line of work is quite promising!--I simply think it is premature, and would be *greatly* improved by analysis of more datasets.

---

### Decision · Program_Chairs · 2019-12-19

**Decision:**

Reject

**Comment:**

The paper examines the idea that real world data is highly structured / lies on a low-dimensional manifold. The authors show differences in neural network dynamics when trained on structured (MNIST) vs. unstructured datasets (random), and show that "structure" can be captured by their new "hidden manifold" generative model that explicitly considers some low-dimensional manifold.

The reviewers perceived a lack of actionable insights following the paper, since in general these ideas are known, and for MNIST to be a limited dataset, despite finding the paper generally clear and correct.

Following the discussion, I must recommend rejection at this time, but highly encourage the authors to take the insights developed in the paper a bit further and submit to another venue. E.g. trying to improve our algorithms by considering the inductive bias of structure of the hidden manifold, or developing a systematic and quantifiable notion of structure for many different datasets that correlate with difficulty of training would both be great contributions.